# Multiplexed Genome Editing for Efficient Phenotypic Screening in Zebrafish

**DOI:** 10.3390/vetsci9020092

**Published:** 2022-02-19

**Authors:** Shuyu Guo, Ge Gao, Cuizhen Zhang, Gang Peng

**Affiliations:** State Key Laboratory of Medical Neurobiology, Ministry of Education Frontiers Center for Brain Science, and Institution of Brain Science, Fudan University, Shanghai 200032, China; 19211520025@fudan.edu.cn (S.G.); gaoge930@163.com (G.G.); czzhang@fudan.edu.cn (C.Z.)

**Keywords:** CRISPR-Cas9, multiplexed genome editing, zebrafish, *tmem183a*, hearing loss

## Abstract

Zebrafish are widely used to investigate candidate genes for human diseases. While the emergence of CRISPR-Cas9 technology has revolutionized gene editing, the use of individual guide RNAs limits the efficiency and application of this technology in functional genetics research. Multiplexed genome editing significantly enhances the efficiency and scope of gene editing. Herein, we describe an efficient multiplexed genome editing strategy to generate zebrafish mutants. Following behavioural tests and histological examination, we identified one new candidate gene (*tmem183a*) for hearing loss. This study provides a robust genetic platform to quickly obtain zebrafish mutants and to identify candidate genes by phenotypic readouts.

## 1. Introduction

With the development of whole genome sequencing (WGS), whole exome sequencing (WES), and genome-wide association studies (GWAS), genetic variants in patients with human genetic diseases can be rapidly identified [1]. Recent advances in sequencing technologies are increasingly being used in medical genetic research and clinical practice [2]. However, sequencing is not sufficient to distinguish true genetic variants associated with disease from broad functional variants in the human genome, and may even generate false positives [3]. This hinders the translation from genetic research to proof of clinical diagnosis, and even potentially limits the biological understanding of the disease. Autosomal recessive diseases account for a large proportion of human genetic disorders. So far, more than 1800 genes associated with autosomal recessive disorders have been identified, but a large number of autosomal recessive genes with recognisable phenotypes still remain to be identified [3]. In addition, 5% of the human genome is still without any functional annotation [4]. These genes are referred to as “dark matter genes” and there is a pressing need for functional validation. Therefore, it is particularly important to refine and expand the genetic, behavioural, and molecular toolkit in animal models.

The zebrafish is a popular model for genetic screening in neuroscience [5]. In zebrafish, the key limiting factors for generating mutant zebrafish are time and animal housing space. Zinc finger nucleases (ZFN) and transcription activator-like effector nucleases (TALENs) are not suitable for high-throughput mutagenesis due to the inefficient assembly of DNA binding domains. CRISPR-Cas9 technology has evolutionized the ability to generate zebrafish mutants [6] and is already widely used in zebrafish due to its simple design, ease of operation, and strong targeting. Unfortunately, the use of individual gRNAs has limited its efficiency and application in biotechnology. As a result, an increasing number of studies have turned to multiplex genome editing instead of individual gRNA synthesis, which has been successfully applied in multiple species [7]. However, a simple and efficient multiplexed genome editing method has not yet been widely used to establish stable homozygous mutants for phenotypic screening, which would allow for the rapid screening of multiple candidate genes from GWAS or WGS, as well as dark matter genes. In this study, we obtained individual homozygous zebrafish mutants in less time, with less housing room, and at a lower cost. We also combined multiplexed genome editing with functional behavioural tests and further molecular biology readouts to identify genes important for hearing loss. By screening for C–start response and then confirming with AMI—43 staining, we identified *tmem183a* mutants showing an obvious hearing loss phenotype, which suggests that *tmem183a* is required for hearing. Overall, this study provides an experimental framework that combines multiplexed genome editing with phenotypic events to perform medium-scale genetic studies in zebrafish.

## 2. Materials and Methods

### 2.1. Experimental Animals

This study was approved by the Ethics Committee of the Shanghai Medical College, Fudan University (approval #20150119088). The wild-type zebrafish used in the laboratory were the AB strain. Mutant zebrafish and AB zebrafish were all raised under consistent conditions. The system water temperature was maintained at 28 °C and pH was maintained at 7.2. The zebrafish were maintained using a 14 h/10 h light/dark cycle: 14 h light (08:00–22:00), 10 h dark (22:00–08:00).

### 2.2. Synthesis of Cas9 mRNA

Plasmids carrying Cas9, as previously described [8], were digested using the XbaI enzyme (New England Biolabs) and incubated for 3 h at 37 °C. The digested plasmids were recovered and purified. In vitro transcription was performed using the T7 high yield RNA Synthesis Kit (New England Biolabs), followed by RNA purification with the GeneJET RNA Purification Kit (Thermo). After purification, the 5’ end of the synthesized Cas9 mRNA was capped using the Vaccinia Capping System (New England Biolabs). The concentration of the capped product was determined after purification and stored at −80 °C for use.

### 2.3. CRISPR-Cas9 Target Selection, Design, and Synthesis

To design gRNA target sequences, the exon sequence of the gene was uploaded onto the website http://skl.scau.edu.cn/targetdesign/ (access on 30 December 2019), then the target gRNA sequences starting with “GG” and ending with “NGG” were selected.

We used a de novo cloning-free method to generate gRNA templates. The template contained the T7 promotor, a gene-specific target sequence, and a common gRNA scaffold (5′-AATTTAATACGACTCACTATAGGN18NGGGTTTTAGAGCTAGAAATAGC AAGTTAAAATAAGGCTAGTCCGTTATCAACTTGAAAAAGTGGCACCGAG-TCGGTGCTTTTT-3′). Six short synthetic oligonucleotides (38–46 nt) were annealed to three fragments, then ligated together to generate the final template. The ligation products were recovered by gel electrophoresis and purified with a GeneJET Gel Extraction Kit (ThermoFisher, Scientific, Rodano, Italy). Then, a T7 high yield RNA synthesis kit (New England Biolabs, Ipswich, MA, USA) was used for in vitro transcription. RNA was purified using a GeneJET RNA Purification Kit (ThermoFisher, Scientific, Rodano, Italy) and stored at −80 °C.

### 2.4. Embryo Injection

Injections were performed in 1–2 cell stage embryos of wild-type zebrafish from the AB strain. Cas9 was diluted and mixed with the gRNAs of five genes on ice. The final concentration was 750 ng/µL for Cas9 mRNA and 30 ng/µL for all five gRNAs. The injection volume was adjusted to approximately 1.5 nL per embryo.

### 2.5. Genotype Identification

A small part of tissue was cut from the tail of the zebrafish to be identified, and the genome was extracted via the alkali lysis method. Genomic targets were identified using PCR amplification reaction, followed by Sanger sequencing. The primers used for PCR amplification were synthesised by Biosune Biotech Co., Ltd. (Shanghai, China) (Table 1).

### 2.6. C-start Response

A 70 mm long, 50 mm wide, 2 mm deep acrylic transparent plate with 16 wells was placed on a vibrating stage. A high-speed camera (Point Grey, Richmond, BC, Canada) was used to record the C-start response and images were recorded at 200 frames/s. The whole apparatus was placed on a white light screen, which was used to achieve visual contrast during recording. The larva at 5 dpf were put into individual wells with 150 µL embryo medium. An oscillator positioned on the plate, which was controlled by the computer, was used to deliver a vibrational stimulus. The fish were allowed to adapt to the wells for 5 min prior to the start of the test. The controller delivered 10 vibrational pulses (200 Hz, 50 ms) at random times within a 10 min interval to minimize the adaptation of the zebrafish. The video sequence was inspected by person blind to the genotype condition, and C-start responses were recorded when the zebrafish began the distinctive C-bend within 25 ms after the start of the vibration stimulus.

### 2.7. Immunohistochemistry

AM1-43 (Biotium, Fremont, CA, USA) was diluted in low calcium Ringer’s solution (LCR), as previously described [9]. Zebrafish larva at 5 dpf were stained with AMI-43 working solution at a concentration of 0.2 μM for 3 min at room temperature in the dark. Samples were rinsed with LCR solution two times for 10 min, then fixed with 4% PFA, shaken for 2 h at room temperature, and placed at 4 °C overnight.

### 2.8. Statistical Analysis

For AM1-43 imaging, zebrafish samples were embedded in 1.6% low melting agarose, then examined by confocal microscopy. The fluorescence intensity values were calculated by custom-written MATLAB scripts. GraphPad Prism software was used for statistical analysis. *p* < 0.05 was considered statistically significant for the t-test.

## 3. Results

### 3.1. Design of CRISPR-Cas9 Gene Editing Systems in Candidate Genes

Our goal was to rapidly obtain homozygous zebrafish mutants by multiplex genome editing techniques. The candidate genes selected were: *gabbr1a*, *gabbr2*, *necap1*, *tmem183a,* and *zgc103499*. *gabbr1a* [10], *gabbr2* [10,11], and *necap1* [12,13] are epilepsy-associated genes, and *tmem183a* and *zgc103499* are highly expressed in inner hair cells [14]. To maximise the indel effect, we selected gRNA target sites located at the N-terminus end of the coding sequences (Table 2), which were expected to result in premature translation terminations by indels. This approach has been shown to be effective in producing simple loss-of-function alleles [15]. PCR was performed to verify the gRNA target sequences in wild-type zebrafish and to exclude possible single nucleotide polymorphisms (SNPs).

Cas9 is the most common endonuclease used to cut target DNA during gene editing. After it forms a ribonucleoprotein with gRNA, it performs a double-stranded cut at the protospacer adjacent motif (PAM), causing a double-stranded break, and thus enabling gene editing [16]. CRISPR-Cas9 can be programmed to target multiple genes at once, which has been demonstrated in a variety of species [17,18,19]. Previous methods used plasmid-based cloning vectors to generate gRNA templates [20]. However, these methods are time-consuming because multiple cloning manipulations are required. The sequence of gRNA we used, as previously described [21], differs from other gRNAs used [16] in that our gRNA contains additional tracrRNA-derived sequences at the 3′ end. In our method, the gRNA template was generated by six oligonucleotides (38–46 nt) that were annealed and ligated in one step (Figure 1). The overhangs of the annealed oligonucleotides were carefully designed to ensure the precise order of ligation and to suppress circulation of the ligation products. With this method, we could routinely generate multiple gRNA templates within 4 h. Then, in vitro transcription was used to generate gRNA.

### 3.2. Multiplexed Genome Editing Led to Five Gene Disruptions in F_0_ Zebrafish

The overall strategy for generating zebrafish and the subsequent screening is shown in Figure 2. To determine the mutation efficiency of the multiplexed editing, we first injected a mixture of in vitro-synthesized Cas9 mRNA and five gRNAs into one-cell stage embryos. Then, 10 of the injected embryos were randomly picked at 24 h post fertilization (hpf) to produce a genomic DNA mixture. The genomic targets were identified using PCR amplification reaction, then sequenced by the Sanger method. Sequencing results showed all five genes were successfully targeted by the multiplexed genome editing method, and the mutation rates varied by around 30–70% as determined by a quantification program [22].

After verification of the mutation events, F_0_ zebrafish were raised to adulthood and in-crossed to generate F_1_ zebrafish. Again, 10 randomly selected F_1_ embryos were assayed for the presence of indel mutations in the F_1_ generation. Sequencing results showed that indel mutations of all five genes were transmitted to the F_1_ animals.

### 3.3. Distribution of CRISPR-Cas9-Induced Mutations in the Study

F_1_ individuals were raised to sexual maturity and out-crossed with wild-type zebrafish. Six F_1_ zebrafish that produced offspring were processed for fin-clips and genomic DNA preparation, followed by PCR and Sanger sequencings. We then examined the sequencing results and determined the diversity and range of the mutations. The gene mutations of each zebrafish are shown in Table 3. At least one mutation was identified in each gene among the six F_1_ zebrafish. Six zebrafish were positive (100%) for at least one mutation, and the average mutation efficiency was 33% for the five genes.

Analysis of the mutant alleles indicated that multiplexed genome editing caused insertions and deletions, as expected. Of all eight mutation types, 60% of the mutations were small deletions, and 30% were complex mutations containing both insertions and deletions. The remaining mutations were insertional mutations. Since the sequences we targeted were in exons, we further analysed whether deletions or insertions changed the reading frame. We identified seven frameshift mutations (87.5%) and one in-frame deletion (12.5%) (Figure 3). The specific types of frameshift mutations for each gene are shown in Table 4. In summary, the high mutation efficiency of each gene demonstrated the utility of CRISPR/Cas9-based multiplexed genome editing technology in zebrafish.

The six F_1_ zebrafish were retained, and their corresponding progeny (F_2_ heterozygotes) were raised to maturity. For each gene, F_2_ heterozygotes were identified by fin-clip and sequencing. The heterozygotes were then in-crossed to produce homozygous mutant offspring, which were used for subsequent experimental studies.

### 3.4. Phenotypic Analysis in Individual Homozygous Mutants

The main purpose of generating homozygous mutants was to examine the functional consequences of the loss of these five genes. To identify whether disease-associated candidate genes could be screened by the multiplexed mutation strategy, we selected two genes associated with hearing loss: *tmem183a* and *zgc103499*. There are three reasons why we chose hearing loss genes for phenotypic analysis: (1) the zebrafish is an important model for studying hearing function. Its genome shares over 80% similarity with humans, and the structure and physiology of hearing pathways are highly conserved between fish and humans [23]. (2) Zebrafish can be tested rapidly for hearing loss by C-start, a stereotypical behavioural response amenable to high throughput assays [24]. (3) Researchers first identified that the key protein TMIE is important in the mechanoelectrical (MET) channel by using zebrafish [25], and the mechanism of DFNB63 non-syndromic hearing loss was also discovered using zebrafish [26]. These two genes are highly expressed in inner ear hair cells, as identified by RNA-seq data [14]. However, there is currently no evidence suggesting that these two genes are associated with hearing loss.

#### 3.4.1. C-start Test

To test our hypothesis, we compared the C-start response rates at 5 dpf between mutant and wild-type zebrafish. The response rate at 5 dpf in wild-type zebrafish, *tmem183a* homozygous mutants, and *zgc103499* homozygous mutants were 68.54%, 43.54%, and 66.88%, respectively. The observed phenotype for *tmem183a* mutants showed a significant decrease in C-start response rate compared to control zebrafish, suggesting that *tmem183* mutants have hearing defects (Figure 4). However, there was no significant difference in the C-start response between *zgc103499* mutants and wild-type zebrafish. Therefore, we repeated the test at 8 dpf, resulting in response rates of 70.31% for control zebrafish and 61.97% for *zgc103499*-KO zebrafish. Again, there was no obvious hearing loss phenotype.

#### 3.4.2. AM1-43 Staining

To further validate the phenotype of *tmem183a* mutants, we investigated their hair cell function. The mechanoelectrical transduction (MET) channel plays an important role in transducing the mechanical force evoked by sound into an electrical signal, which is responsible for hearing. Mutations in TMC1 cause hearing loss by disrupting mechanoelectrical transduction [27]. In zebrafish, hair cells are located both in the inner ear and the lateral line. The lateral line consists of clusters of hair cells arranged sequentially along the body from head to tail; these are more accessible for experimental operation and microscopic imaging [28]. AM1-43 is a live fluorescent dye that can enter the hair cell through the MET channel, and the intensity of intracellular staining is used to detect the state of the MET channel [29]. We found that the intensity of AM1-43 staining in the hair cells of *tmem183a* mutant animals was significantly reduced to 64.66% of the control at 5 dpf. To confirm the hair cell reduction in *tmem183a* mutants, we also analysed the number of hair cells in lateral line neuromasts. In wild-type zebrafish at 5 dpf, there was an average of 10.4 hair cell clusters per neuromast, but this was significantly decreased to 7.34 in *tmem183a* mutants (Figure 5). Hence, the *tmem183a* gene mutation may affect the normal state of MET channels in hair cells, which may result in a decline in hearing function. These results demonstrated that multiplexed genome editing could be efficiently used to screen for hearing loss defects in zebrafish mutants.

## 4. Discussion

The zebrafish is an excellent model for genetic disease, as it can rapidly provide qualitative and quantitative data relating to both genotypes and phenotypes. Therefore, unlike most other vertebrates, zebrafish are appropriate for high-throughput screening [30]. CRISPR-Cas9 gene editing technology has greatly improved the ability to accurately identify and functionally validate the genes and molecular mechanisms responsible for genetic diseases [31]. The traditional construction of multiple homozygous zebrafish mutants is time-consuming and inefficient. Multiplex genome editing technology, a reliable gene editing tool developed in recent years, has greatly enhanced the scope and efficiency of gene editing [32]. Herein, a novel strategy is proposed for multiplex genome editing that can simultaneously target five genes and rapidly obtain homozygous mutants. Based on this high-throughput protocol, homozygous mutants can be obtained in a shorter time and markedly fewer animals are used.

Most methods for assembling gRNA are derivatives of cloning methods, such as oligo assembly, PCR, and Golden Gate [7]. These methods have many complicated cloning steps and are time-consuming. The gRNA assembly described in this study is simple and efficient, resulting in six annealed oligos containing 5’ and 3’ overhangs on either end to ensure a precise order of assembly. This method saves the cost of cloning and sequencing and shortens the time required to obtain gRNA templates. In addition, this approach is simpler in the context of a multiple-gene editing strategy. During the sequence design process, only the target sequence of the gene needs to be replaced in two sequences, and the remaining fragments do not need to be resynthesized.

The average mutation efficiency of the five genes obtained by our multiplexed genome editing approach was 33%, which is similar to the mutagenesis efficiency of single genes in zebrafish [8]. Similarly, the average mutagenesis efficiency of genes in human cells by CRISPR-Cas9 technology targeting 10 genes simultaneously was similar to the mutagenesis efficiency of single genes [33]. In yeast, targeting eight genes simultaneously using multiple gRNAs has been achieved, and up to 86.7% of cells carried mutations in all eight genes [34]. Previously, Jao et al. reported that five genes can be disrupted simultaneously in zebrafish. They observed a combination of phenotypes in single F0 embryos after injecting the embryos with a mixture of five gRNAs and Cas9 RNA [35]. We generated individual homozygous mutants for five genes, and in addition, we observed germ-line transmission of the mutations. Varshney and colleagues reported a cloning-free method to generate gRNA templates, and they also successfully carried out a multiplexed genome editing trial against 10 genes. They used two partially overlapping oligonucleotides (60 nt and 80 nt) and Taq DNA polymerase-mediated DNA synthesis to generate the gRNA templates [36]. The cloning-free method we used in this study has its root in our previous work of de novo synthesis of TALE repeats [37]. Thus, in our design, we have divided the final gRNA template into three fragments and used short oligonucleotides of around 40 nt to minimize the mutations introduced by synthetic errors in chemically synthesized oligonucleotides. However, it seems unlikely that synthetic errors in oligonucleotides are a key obstacle in the synthesis of the gRNA template, as the final template is only about 120 bp and significantly smaller than the TALE arrays. We have also simultaneously targeted five genes. As the number of gRNAs in a cell increases, they must compete for the dwindling ‘pool’ of endonucleases. This competition leads to decreased mutation efficiency, which is called retroactivity [38]. Therefore, the design of our strategy was to target five genes in zebrafish, which takes into account the burden on the 1–2 cell stage of zebrafish embryos, and also avoids the inefficiencies caused by endonuclease competition, while still ensuring throughput.

In the study of gene function in zebrafish mutants, single gene mutations are typically achieved by the injection of a single gRNA into zebrafish embryos, followed by time-consuming, labour-intensive genetic hybridisation and subsequent genotyping. A traditional knockout strategy that seeks to simultaneously perform single gene knockouts for five genes requires the identification of nearly five times more zebrafish than our approach. When the number of zebrafish increases, not only are there higher housing costs, but maintaining their environment also becomes increasingly challenging [39]. Our strategy can also save up to 80% more space in fish housing. If space constraints necessitate the five genes to be sequentially knocked out at different periods, our approach can help experimenters save 2–3 years.

The limitations of the proposed method for rapid access to homozygous mutants by multiple knockouts in this paper are the potential off-target effects and the complexity of the genetic background. As intracellular gRNA increases, the off-target effect is enhanced accordingly [7]. Several online toolkits have been developed that can be used to predict off-target rates, in order to reduce the possibility that off-targeting occurs [40,41]. Recent studies have also reported reduced off-target effects by combining a modified, ineffective Cas9 endonuclease with an engineered reverse transcriptase to perform gene editing without DNA double-strand breaks [42]. Additional rounds of out-crosses may be employed to minimize the effects of likely off-target events.

There is a strong need for medium- or large-scale genome-editing projects to help determine the consequences associated with the loss of function of genes of interest. Therefore, we combined behavioural tests and molecular biology methods to perform rapid phenotypic screening in zebrafish. We selected two dark matter genes which may play a role in hair cell development. *tmem183a* mutations caused changes in C-start response rate while *zgc103499* did not, so we further investigated *tmem183a* mutants with AM1-43 staining to determine hair cell function. Our results showed that the AM1-43 staining of *tmem183a*-KO zebrafish was abnormal, consistent with the conclusion that this gene may be associated with hearing loss. Several studies have also recently reported rapid phenotypic screening by CRISPR-Cas9 technology. By simultaneously injecting multiple gRNAs of a single gene into wild-type zebrafish embryos, researchers can rapidly screen for cardiovascular phenotypes in the F_0_ generation [43]. Although this approach has a time advantage, it has a strong off-target effect, and it is difficult to achieve multiple-gene screens. We demonstrated that multiple gRNAs co-injected with Cas9 RNA into zebrafish embryos can produce mutations in multiple target genes, mostly frameshift mutations that can lead to loss of gene function, and that these mutations are independently inherited in the germ-line to produce homozygous offspring.

## 5. Conclusions

The zebrafish is a leading animal model for studying human genetic diseases. Using a functional C-start assay to pre-screen for phenotypes prior to AM1-43 staining validation, we identified one new gene (*tmem183a*) associated with hearing loss. Hence, reliable multiplexed genome editing was combined with functional phenotyping and histology to produce a powerful approach for hearing loss screening in zebrafish. This strategy can also be applied to screen for other disease candidate genes. This research provides a valuable platform to conduct medium- or large-scale screening and validation of the functions of important genes.

## Figures and Tables

**Figure 1 vetsci-09-00092-f001:**
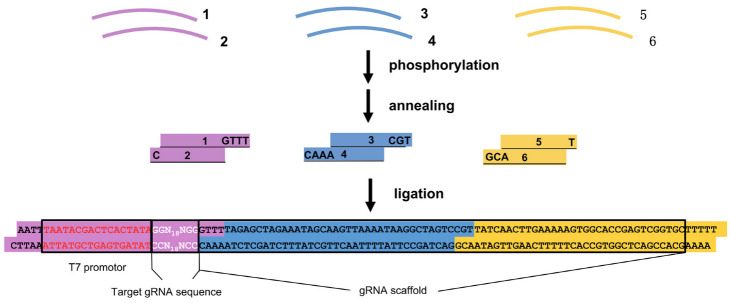
A schematic diagram of generating gRNA templates. Each template contains the T7 promotor, a gene-specific gRNA target, and a common gRNA scaffold. The template is synthesized by ligating three fragments together (in purple, blue, and yellow color), which are annealed from six chemically synthesised oligonucleotides (1–6). The sequences of the overhangs of these three annealed fragments are shown. White letters in the purple fragment designate the gRNA target sequence.

**Figure 2 vetsci-09-00092-f002:**
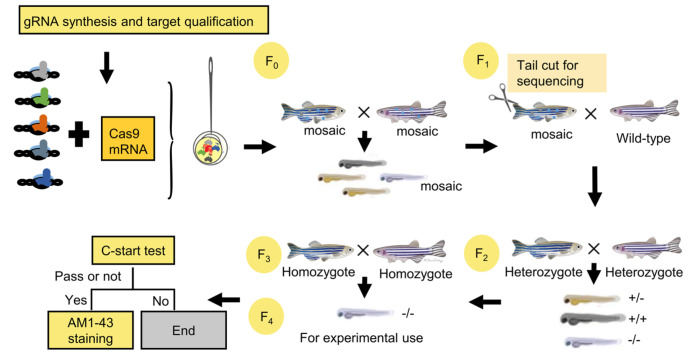
An overview of multiple-gene knockout technology and the generation of individual homozygous mutants. Cas9 mRNA and five gRNAs were synthesized and co-injected into 1–2 cell stage embryos of wild-type zebrafish. Mosaic F_0_ were in-crossed to generate mosaic F_1_ zebrafish. Then, F_1_ fish were out-crossed with wild-type to generate F_2_ fish. The gene mutation results of mosaic F_1_ were identified by sequencing. Heterozygous F_3_ zebrafish were crossed to generate homozygous mutants.

**Figure 3 vetsci-09-00092-f003:**
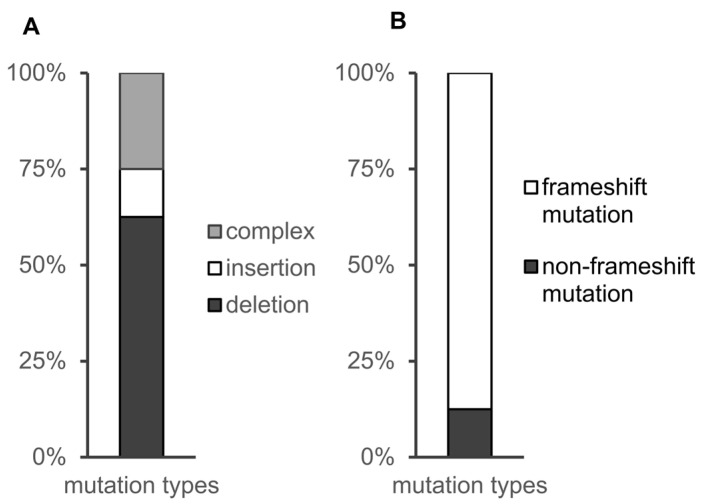
The mutations were classified into different types. (**A**) Comparison of the types of mutations. The mutations were classified in three different groups: deletions, insertions, and complex (at least one deletion and at least one insertion). (**B**) Comparison of the types of mutations detected by sequencing. The mutations were classified in two different groups: frameshift mutation(change the reading frame) and non-frameshift mutation.

**Figure 4 vetsci-09-00092-f004:**
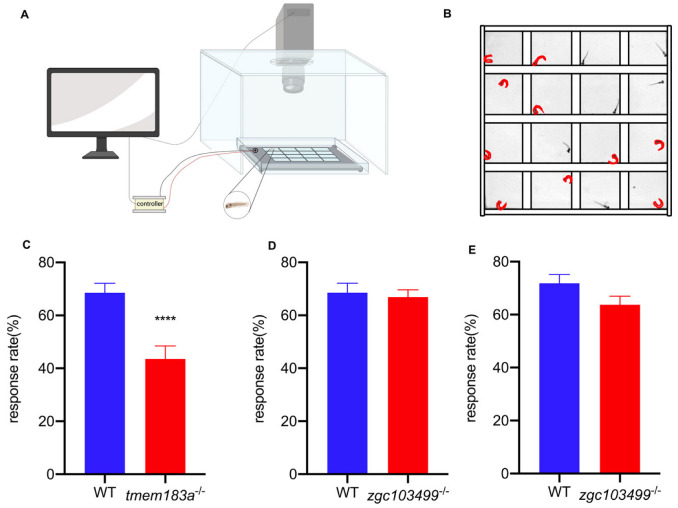
C-start response in *tmem183a*- and *zgc103499*-deficient zebrafish larva: (**A**) a diagram of the C-start experimental apparatus. A transparent plastic plate was placed on top of a light screen, which was vibrated by an oscillator. Each well was filled with 150 µL embryo medium and one zebrafish larva. The responses of 16 zebrafish were recorded in one session; (**B**) the representative frame of the C-start response in zebrafish; (**C**) the mean response rate of *tmem183a*-deficient zebrafish at 5 dpf. N(wild-type) = 48, N(*tmem183a*^-/-^) = 48, ****: *p* < 0.0001; (**D**) the mean response rate of *zgc103499*^-/-^ zebrafish at 5 dpf. N(wild-type) = 48, N(*zgc103499^-/-^*) = 48, ns; (**E**) the mean response rate in *zgc103499*^-/-^zebrafish at 8 dpf. N(wild-type) = 48, N(*zgc103499*^-/-^) = 48, ns.

**Figure 5 vetsci-09-00092-f005:**
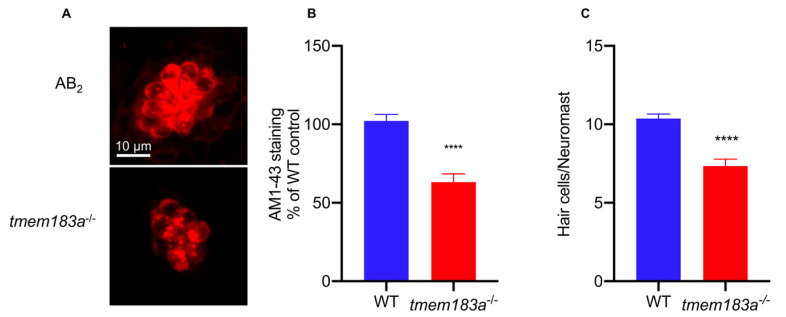
The tmem183a mutation impaired the hair cells of zebrafish: (**A**) confocal microscopic analysis of the hair cells in wild-type and *tmem183a*-KO zebrafish at 5 dpf; (**B**) quantification of the staining intensity of hair cell clusters at 5 dpf (for triplicate experiments). N(wild-type) = 32, N(*tmem183a* mutant) = 35, ****: *p* < 0.0001; (**C**) quantification of hair cell clusters at 5 dpf. N(wild-type) = 33, N(*tmem183a* mutant) = 38, ****: *p* < 0.0001.

**Table 1 vetsci-09-00092-t001:** Primer information for five genes.

Gene	Forward (5′-3′)	Reverse (5′-3′)
*gabbr1a*	CTTGGTGTTGTGGTAGTAGCATTACA	GGGATGTTATGTGTTTATGTCTTACC
*gabbr2*	GTGGATCTCAGTTGTTCGGTAGC	GTAGATTTTGGTGATGAGTGAGAAGG
*necap1*	ACCATCACCAAAAAGTGGTATGGCA	CCTGAAGACAGTTATGAAGCTCACGT
*tmem183a*	ACCTTCCTGATCGTCTTCAGCCT	ATTCCCACATCTGCTCTGACCTAC
*zgc103499*	GAGACACAGAGAGAGAAATTGTTAGAAG	GAGTTTGAGAACCCCCGCTTTAGA

**Table 2 vetsci-09-00092-t002:** The target gRNA sequences of five genes.

Gene	Forward (5′-3′)
*gabbr1a*	GGATGTCCCTTGAGAACGGGAGG
*gabbr2*	GGCACGGCCTGGACAACAACTGG
*necap1*	GGAAGTTGGACGCTCCTGACTGG
*tmem183a*	GGTGTAGATTCGGGGAGAGGAGG
*zgc103499*	GGATCGGAAACCCACCAAGCAGG

**Table 3 vetsci-09-00092-t003:** A summary of the mutant identification results of F_1_ zebrafish. × indicates that the mutant has no mutation in the corresponding gene.

Gene	Penta KO #1	Penta KO #2	Penta KO #3	Penta KO #4	Penta KO #5	Penta KO #6	Mutation Efficiency
*gabbr1a*	+2	×	+2	×	×	−8	50%
	(−2, +4)		(−2, +4)				
*gabbr2*	×	×	+1	×	×	×	16.67%
			(−2, +3)				
*necap1*	×	−4	×	−10	×	×	33%
*tmem183a*	−5	×	×	×	−5	×	33%
*zgc103499*	×	−33	×	×	×	+1	33%

**Table 4 vetsci-09-00092-t004:** The frameshift mutation types of different genes. For each gene, the wild-type sequence is shown at the top with target sites. Deletions are shown as dashes and insertions highlighted in blue.

Gene	Specific Sequence	Frameshift Mutation Types
*gabbr1a*	GGATGTCCCTTGAGAACGGGAGGGTGTCGCTG	Wild-type
	GGATGTCCCTTGAG--------GGTGTCGCTG	−8
	GGATGTCCCTTGAGAA-----GGAGGGTGGGTGTC	+2 (−5, +7)
*gabbr2*	GGCACGGCCTGGACAACAACTGGTACGCG	Wild-type
	GGCACGGCCTGGACA---CGCGACTGGTACG	+1 (−3, +4)
*necap1*	TCCAGTCAGGAGCGTCCAACTTCCAGTCCG	Wild-type
	TCC----------GTCCAACTTCCAGTCCG	−10
	TCC----AGGAGCGTCCAACTTCCAGTCCG	−4
*tmem183a*	CCTCCTCTCCCCGAATCTACACCCACT	Wild-type
	CCTCC-----CCGAATCTACACCCACT	−5
*zgc103499*	CCTGCTTGGTGGGTTTCCGATCCAGG	Wild-type
	CCTGCTATGGTGGGTTTCCGATCCAGG	+1

## Data Availability

Not applicable.

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
