# Peer review of "Multiplexed Genome Editing for Efficient Phenotypic Screening in Zebrafish"

_vetsci, 2022, doi:10.3390/vetsci9020092_

Round 1

Reviewer 1 Report

This works uses a cloning-free method to perform multiplexed genome editing in zebrafish. The authors tested the efficiency of simultaneously targeting 5 genes and examined mutant phenotypes. Similar methods have been reported previously, so the novelty of the present work is limited.

1. Multiplexed genome editing and cloning-free sgRNA synthesis have been reported (Jao et al., 2013; Varshney et al., 2015). The authors should clearly explain and discuss the novelty of this work with respect to previous studies.

2. Figure 1 is not clear.  Authors need to provide more explanation in the legend. What the three fragments correspond to? What the “A” under olio 4 means?

3. What are the advantages of the “six oligonucleotides” method for generating sgRNA compared with previously reported cloning-free sgRNA synthesis? For example Varshney et al., 2015

4. Relevant works published by others were not discussed in the manuscript.

5. Table 4, the inserted base in zgc103499 should be highlighted.

6. By examining mutant phenotypes, authors found defective hair cell development following tmem183a knockout. This is interesting but it is not sufficiently characterized. In Figure 5, authors at least need to show whole embryos stained by AM1-43 and examine the formation of neuromasts.

7. Line 15, “quickly rapidly and effectively” should be ““quickly, rapidly and effectively”.

8. Line 29, “The With the development…” should be “With the development…”.

9. Lines 151-152, citation of “Wang et al., 2016” should follow the journal style. 

Reviewer 2 Report

Guo et al presented a CRISPR-Cas9-based multiplexed genome editing method in zebrafish and used this method to generate mutants in multiple selected genes and investigate a candidate gene for hearing loss phenotype. The method is relevant to the field, and the manuscript is presented well although with some needs for improvement. See comments below:

1) The Methods section need more details, especially considering this is a method-focusing paper. For example, which Cas9 plasmid is used (as there have been multiple gene editing systems so far); is the gRNA transcribed the same way as Cas9 mRNA; how is the genotype determined; etc.

2) Fig 1 needs to describe the fragment of each color in text.

3) Although the multiplexed method could indeed save animals and time as stated in the Discussion, one potential disadvantage of the proposed method compared with the traditional single-gene knockout method is that possibly more than one gene mutations are generated in the same cell, yielding animals with different mutations inherited together instead of independently. In the Results section, the authors may need to present data showing whether the homozygous mutant offsprings they tested are all single-gene mutants or also include some multi-gene mutant cases.

4) Grammar: L29 The; L268 due to --> as; etc.
